# Biological Factors Underpinning Suicidal Behaviour: An Update

**DOI:** 10.3390/brainsci13030505

**Published:** 2023-03-16

**Authors:** Maya N. Abou Chahla, Mahmoud I. Khalil, Stefano Comai, Lena Brundin, Sophie Erhardt, Gilles J. Guillemin

**Affiliations:** 1Department of Biological Sciences, Faculty of Science, Beirut Arab University, Beirut 11072809, Lebanon; 2Molecular Biology Unit, Department of Zoology, Faculty of Science, Alexandria University, Alexandria 21568, Egypt; 3Department of Pharmaceutical and Pharmacological Sciences, University of Padua, 35131 Padua, Italy; 4Department of Biomedical Sciences, University of Padua, 35131 Padua, Italy; 5Department of Psychiatry, McGill University, Montreal, QC H3A1A1, Canada; 6Division of Psychiatry & Behavioral Medicine, Michigan State University College of Human Medicine, Grand Rapids, MI 49503, USA; 7Center for Neurodegenerative Science, Van Andel Research Institute, Grand Rapids, MI 49503, USA; 8Department of Physiology and Pharmacology, Karolinska Institutet, 17177 Stockholm, Sweden; 9Neuroinflammation Group, Faculty of Medicine, Macquarie University, Sydney, NSW 2109, Australia

**Keywords:** suicide, suicide attempts, suicidal ideation, neuroinflammation, epigenetics, hypothalamic-pituitary-adrenal axis, kynurenine pathway, risk factors, psychiatric disorders

## Abstract

Suicide, a global health burden, represents the 17th leading cause of death worldwide (1.3%), but the 4th among young people aged between 15 and 29 years of age, according to World Health Organization (WHO), 2019. Suicidal behaviour is a complex, multi-factorial, polygenic and independent mental health problem caused by a combination of alterations and dysfunctions of several biological pathways and disruption of normal mechanisms in brain regions that remain poorly understood and need further investigation to be deciphered. Suicide complexity and unpredictability gained international interest as a field of research. Several studies have been conducted at the neuropathological, inflammatory, genetic, and molecular levels to uncover the triggers behind suicidal behaviour and develop convenient and effective therapeutic or at least preventive procedures. This review aims to summarise and focus on current knowledge of diverse biological pathways involved in the neurobiology of suicidal behaviour, and briefly highlights future potential therapeutic pathways to prevent or even treat this significant public health problem.

## 1. Introduction

Suicidal behaviour (SB) has gained an international attention at the social, economic and scientific levels. Suicidality is interpersonally and situationally complex, and an in-depth study into its basic causes has become a compelling field of interest [1]. Suicidal thoughts and behaviours (STBs) include suicidal ideation (SI), preparatory behaviours, suicide attempts (SA), and suicide. SI is the desire to end one’s own life; SA is a non-fatal behaviour with the intent to die [2]. The death due to a fatal behaviour caused by an intentional and voluntary injury directed towards one’s self with the aim to die is suicide [3]. About 90% of suicidal samples have an underlying mental/psychiatric disorder such as major depressive disorder (MDD) or bipolar disorder (BP) [4]. The risk of suicide is 20-fold higher in MDD patients compared to the entire community [5]. Furthermore, BP patients with comorbid anxiety, showed increased SAs rate compared to controls [6]. Annually, more than 700,000 individuals, or one person every 40 s die by suicide worldwide [7], but SAs occur 10–20 times more often [8]. The reported statistical number of deaths due to suicide is underestimated, as a result of ethical/religious perspectives or even death-cause misclassification. Suicidality, indeed, has severe economic/social consequences [9]. Thus, suicide is a complex/multi-factorial phenomenon, often associated with mental disorders and/or other non-psychiatric traits. Examining suicidal phenotypes inside and outside psychopathology is crucial to elucidate risk factors’ nature, predict and prevent suicide occurrence [10].

Suicide risk factors consist of cultural, economic (influence nutrition), psychological/psychiatric, psychosocial, and biological factors [8]. Psychosocial factors are of great importance to mention here and they can be defined as the social structures/processes which can intervene with individual’s health/behavioural outcomes and thoughts affecting mental health and representing the foundation of a suicidal character such as adverse life events (sexual/physical abuse), family discord/violence, peer conflicts, lack of communication and support, excessive work demands, personality traits such aggression/impulsivity and neuroticism, perfectionism, pessimism, interpersonal dependency, low self-esteem, social isolation and discrimination [11,12,13]. Other factors that increase stress and hopelessness and sequentially contribute to SB are peri-partum depression in pregnant women [14], alcohol related self-harm (Self-harm can be suicidal or non-suicidal; suicidal self-harm requires intention in addition to the injurious act) that accounts for 10–15% of suicidal deaths [15], chronic diseases (cancer, HIV) [7], severe stress [16], smoking (nicotine-induced activation of microglial cells), substance abuse, low serotonin levels and irregular specific cytokines levels [17]. Regulators of suicidal risk include behavioural/interpersonal traits, coping mechanisms, lifestyle and psychiatric diagnosis (MDD, BP, schizophrenia) [18]. Disorders such as post-traumatic stress disorder, borderline-personality disorder, antisocial-personality disorder, anorexia-nervosa and sleep disorders—and negative traits such as hopelessness and impaired cognitive functions—increase suicidal risk. Regarding suicide method lethality, a meta-analysis of 34 studies assessing the case fatality rate (CFR %) of various suicide methods has provided evidence about the lethality of the method based on the act used, where firearms was classified as the fastest, most lethal and most often an irreversible act (~89.7%), followed by suffocation/hanging (~84.5%), drowning (~80.4%), gas poisoning (~56.6%), jumping (~46.7%), liquid or drug poisoning (~8.0%) and cutting (~4.0%). Changes in lethality might vary overtime according to population of study and age groups [19,20].

Epidemiological studies showed that completed suicide in males is three times more common than in females; whereas SAs are more prevalent in females [21], but this data is susceptible to variation at the level of countries and other factors including social, cultural and economic. Importantly, suicidal risk factors are age-dependent, and stress represents the proximal risk factor controlling time/probability of SB, while diathesis (vulnerability) is the distal risk component affecting SB independent of the psychiatric condition [22]. To understand SB, many theoretical models and scales were established, including cognitive/interpersonal theories, biological/psychodynamic theories of suicide and the diathesis-stress model. Physiological tools to evaluate the risk of suicidal behaviour include: (1) Beck Suicide Intent Scale (SIS), (2) Beck Scale for Suicide Ideation (SSI), (3) Suicide Assessment Scale, (4) Beck Hopelessness Scale, (5) Karolinska Interpersonal Violence Scale, (6) Columbia Suicide Severity Rating Scale, (7) Harcavy-Asnis Suicide Scale, and (8) Lethality of Suicide Attempt Rating Scale (LSARS-II). In children, scales used to evaluate the risk of suicide include: (1) Child Suicide Potential Scale, (2) Evaluation of Suicide Risk Among Adolescents, and (3) Imminent Danger Assessment [23]. Even though animal models are validated to study the most important risk factors of suicide [24], the hardship of an accurate suicide risk determination and the lack of animal models that closely resemble human SB are the greatest obstacles against understanding the neuroscience of suicide. However, some components involved in the neurobiology of SB, and circuits relevant to ideation in humans and some pharmacological quantitative aspects, can be studied in proper animal models such as rodents [25]. SB is a phenomenon and disorder of growing importance and high impact on the population, thus was classified by the *Diagnostic and Statistical Manual of Mental Disorders: Fifth Edition* as a condition for further study [26]. Here, we aimed to focus on reviewing evidence about various biological factors involved directly or indirectly in the pathophysiology of SB. Understanding the biological factors, mechanisms and alterations strengthen the detailed knowledge of SB and its phenotypes and deepen the link among interconnected/involved pathways which will in turn advantageously influence the therapeutics of SB. For this purpose, we performed an advanced search on recent research concerned with SB through PubMed, PubMed Central, ScienceDirect and Google Scholar databases.

## 2. Biological Alterations Underlying Suicidal Behaviour

A substantial number of studies demonstrated a robust combination of mechanisms and pathways (Figure 1) that have a direct impact on or might be a causative factor for SB.

### 2.1. Genetic/Molecular Factors

#### 2.1.1. Genome-Wide Association Studies

Genome wide association studies (GWAS) indicated that SB is polygenic and genetics contribute to more than 43% of SB [27], empowered by studies assessing twins, adoption and immigrant populations [25]. In addition, gene-environment-interaction findings explain the high co-occurrence/co-morbidity between psychiatric disorders and SB. However, although individuals with mental disorders are at higher risk to develop SB, SB is based on a complicated genetic background that varies from that of mental disorders [28]. To date, more than 2500 genes have been identified to be associated with suicide, and 40 of them are related to cell cycle and DNA repair, but require further statistical examination [29]. GWAS on a large European ancestry population identified two genome-wide significant loci involving six single-nucleotide polymorphisms (SNPs) (*rs34399104*, *RS35518298*, *rs34053895*, *rs66828456*, *rs35502061*, and *rs35256367*) with 25% significant SNP-based heritability of suicide and implication of 22 genes on the following chromosomes: 13, 15, 16, 17, and 19. Another GWAS to identify genetic variants associated with broadly-defined suicidality in 122,935 participants of the UK Biobank cohort—and to determine whether increased genetic burden for suicidality was associated with both psychiatric disorders and completed suicide in a non-overlapping sample—has led to the identification of three novel genome-wide significant loci for suicidality (on chromosomes 9, 11 and 13 having the following SNPs *rs62535711*, *rs598046*, *rs7989250*, respectively) [30,31]. In Denmark alone, a huge study involving 77639 individuals from the general population identified numerous loci associated with SAs when adjusting with socio-demographics (Sex, Age range, Living arrangements, Area Deprivation): *rs6880062* and rs6880461 on Chromosome 5. Another three significant associations with SAs were detected on Chromosome 20, when adjusting for mental disorders: *rs4809706*, *rs4810824*, and *rs6019297* [32]. Conversely, GWAS on the Japanese general population (746 suicides) detected no significant loci, but revealed 35–48% SNP-based heritability and a polygenic inheritance of completed suicide. However, larger sample sizes would be preferable in order to draw robust conclusions about this and to detect reliable genetic markers. This study identified a novel suggestive locus, intronic SNP, in the GTF2I Repeat Domain Containing 1 (*GTF2IRD1*) gene, which is responsible for neurodevelopmental abnormalities and was found to be correlated with age in completed suicide. However, this necessitates replication studies with a larger sample size to confirm this result [33]. In addition, single nucleotide variants and copy number variants are implicated in STBs, whose real influence must be confirmed by replication studies [10].

The genetic architecture of SA was studied through a meta-analysis identifying pan-ancestry and ancestry-specific risk loci proving the complexity of SB. The following pathways—oxytocin signalling, glutamatergic synapse, cortisol synthesis and secretion, dopaminergic synapse, and circadian rhythm—showed over-representation and high clinical significance in suicidal samples [34]. Another GWAS meta-analysis on 29,782 SA cases from 18 cohorts worldwide has identified two significant loci for SA, the major histocompatibility complex and an intergenic locus on chromosome 7 (implicated in risk-taking behaviour, smoking, and insomnia), where the latter is more strongly associated with SA than psychiatric disorders. Significant genetic overlap was also found between SA and many non-psychiatric traits including reproductive traits, smoking, sleep disturbances, lower educational attainment, lower socioeconomic status, poorer overall general health, risk-taking behaviour and pain [35]. In addition, a meta-analysis has been conducted since 2011 and still ongoing on 633,778 US military veterans of different ancestries (African, Asian, European, Hispanic). The meta-analysis has identified cross-ancestry risk loci in the following candidate genes: Estrogen Receptor 1 (*ESR1*), Dopamine Receptor D2 (*DRD2*), TNF Receptor Associated Factor 3 (*TRAF3*) and DCC Netrin 1 Receptor (*DCC*). This will further help in clarifying the molecular basis of STBs, but replication studies are needed to confirm their role in suicidality [36]. In addition, an association between SB and glucose regulation, and SB and protein localization regulation was confirmed by GWAS studies [37].

#### 2.1.2. Gene Expression Alteration and SB

The glutamatergic-signalling-pathway which is partly regulated by sex-related hormones, represents the primary excitatory neurotransmission pathway in the brain—involved in intracellular-calcium-signalling and cell survival and proliferation—is altered in suicidal patients [38]. The leucine rich repeat transmembrane neuronal 4 gene (*LRRTM4*), expressed in neurons of the central nervous system (CNS), has two male-specific variants, five male-specific haplotypes and one female-specific haplotype in its large intron-3, and was found to be associated with SAs. *LRRTM4* encodes the neuronal leucine-rich repeat transmembrane protein that localizes to excitatory synapses to promote glutamatergic synapse development [38]. Cardiac-B-adrenergic signalling is also a suicide-specific pathway [39]. Moreover, the following three genes—SH3 And SYLF Domain Containing 1 (*SH3YL1*), Acid Phosphatase 1 (*ACP1*) and ALK And LTK Ligand 1 (*ALKAL1*)—expressed in the brain, showed five variants in males suicide attempters, confirming sex-associated suicidality [40]. Weighted-gene co-expression-network analysis has identified 10 hub genes highly associated with suicide and incriminated in vital signalling pathways including oestrogen, legionellosis, glucagon signalling pathways, and the activation of nitric-oxide synthase and endoribonuclease [41].

Glial gene expression in the dorsolateral prefrontal cortex (DLPFC) and anterior cingulate cortex (ACC) was assessed to monitor changes in expression of specific markers and their relation to SB. This indicates differential changes in several glial transcripts, especially for microglial markers in suicidal completers where a 20% increase in the CD68 mRNA level in the DLPFC was observed. This in turn explains the increase in cerebral monokines including tumour necrosis factor-α (TNF-α) and interlukin-1β (IL-1β) whose levels in suicide attempters were also higher compared to individuals with SI only. Whereas, there were no significant changes in gene transcripts encoding astrocytic markers or oligodendrocytic markers for suicide attempters [42]. Furthermore, alterations in the cannabinoid receptors (CB_2_r) and the non-cannabinoid receptors (GPR55) were associated with suicide. These receptors are key-targets involved in response to depression, stress/anxiety, and also in controlling the process of decision making [43]. Gene-expression-analysis by real-time PCR of *CB_2_r* and *GPR55* in the DLPFC, a region involved in decision-making and cognitive functions, showed downregulation (33 and 41%, respectively) in 18 suicidal victims with no clinical psychiatric history. However, at the protein-expression level using western-blot, CB_2_r expression was higher with no significant changes of the GPR55 protein expression. In addition, CB_2_-GPR55 hetero-dimer complexes protein expression was higher in suicidal victims in neurons and astrocytes but not in the microglia [43]. The endocannabinoid system implicated in the regulation of emotional response, represents a potential target in suicidal research [44]. Altered gene expression of gamma-aminobutyric acid (GABA)ergic and adenosine triphosphate (ATP) biosynthesis pathways were detected in suicidal completers too [45]. The following eight genes—Interleukin-6 (*IL-6*), Spermidine/spermine N-1 acetyltransferase (*SAT1*), Spindle And Kinetochore Associated Complex Subunit 2 (*SKA2*), *Myelin Basic Protein* (*MBP*), *Jun* Proto-Oncogene, AP-1 Transcription Factor Subunit (*JUN*), Kelch Domain Containing 3 (*KLHDC3*), Kinesin Family Member 2C (*KIF2C*) and Solute Carrier Family 4 Member 4 (*SLC4A4*)—were also considered as top-ranking markers in predicting SI [46]. Moreover, contactin-5 (*CNTN5*), 26S proteasome non-ATPase regulatory subunit 14 (*PSMD14*), hepatic and glial cell adhesion molecule (*HEPACAM*) and hepatocellular Carcinoma, Down-Regulated 1 (*HEPN1*) are four genes associated with suicidality [1]. Changes have also been recognized in potential regulatory genes including Zinc Finger Protein 714 (*ZNF714*), and Nuclear Receptor Interacting Protein 3 (*NRIP3*), in addition to Catechol-O-Methyl-Transferase (*COMT*) in suicidal samples at the gene-expression level due to differences in DNA methylation status [47]. A whole-transcriptome brain expression study revealed that Humanin-like 8 (*MTRNR2L8*) was higher in suicide, in contrast to serpin peptidase inhibitor, and member 1 (*SERPINH1*) [45]. Additionally, DNA-dependent ATPase activity was found to be augmented in suicide [45], and changes in polyamine expression (implicated in immunity, oxidative stress, cell proliferation/apoptosis) and their metabolic enzymes, suggest their possible role in suicidality [48]. Nucleolar Organizing regions are genetic loci composed of ribosomal DNA (rDNA) and proteins. Molecular studies showed that the rDNA transcriptional activity, which plays a key role in neural plasticity of dorsal raphe nucleus (DRN) neurons, was decreased in the hippocampus of suicidal patients compared to controls, regardless of any underlying psychiatric diagnosis [49]. Research also showed the association of the ribonucleic acid (RNA) editing of serotonin receptor 2C (5-HT2C) that would result in receptor isoforms that activate guanine nucleotide-binding protein (G-protein) in suicidal samples. RNA splicing is under examination for its possible impact on suicide [45].

#### 2.1.3. Methylation Status of Some Genes Implicated in SB

Epigenetic modifications include DNA methylation, histone acetylation and non-coding RNAs that orchestrate gene expression without altering DNA sequence. DNA methyltransferases (DNMTs), involved in the addition of a methyl group to DNA nucleotides, were highly expressed and altered in suicidal completers and individuals with psychiatric disorders [39]. Epigenome-wide association studies identified alterations in DNA methylation patterns of genes such as the Glutamate Ionotropic Receptor Kainate Type Subunit 2 (*GRIK2*) and the Spindle and Kinetochore Associated Complex Subunit 2 (*SKA2*) genes, and others involved in pathways linked to neuronal plasticity [39]. Particularly in the PFC, frontal cortex and Wernicke area, a high level of DNA methylation was recognized in suicidal brain samples [18,50]. Suicide-specific analysis found that 55% of differentially-methylated regions (DMRs) were hypermethylated in suicidal samples [39]. The PFC and cerebellum (CER) of suicide completers revealed evidence of altered DNA methylation status compared to controls in both brain regions. Genes in the CER and PFC in the top-ranked DMR, located in intron-1 of the tryptophan rich-basic-protein gene, is hypomethylated on all five CpG sites in suicidal cases. *CERC2* associated DMR (found in the intronic-region of the gene on chromosome 22) showed hypermethylation in the four CpG sites in suicidal cases in addition to DMRs in exon-10 of the solute carrier family 4 member 4 (*SLC4A4* gene), exon-3 of the WW Domain Containing Transcription Regulator 1 (*WWTR1)* gene, and finally in the promoter region of mediator complex subunit 13L (*MED13L*) gene were associated with suicide [18]. The intronic region of the ELAV Like RNA Binding Protein 4 (*ELAVL4*) gene—which plays a significant role in translation/stabilization of mRNA and is a negative regulator of proliferation/differentiation in neural stem cells in the brain maintaining and modulating neuronal development—showed consistent hypomethylation in suicidal cases [18]. Numerous candidate genes including Brain Derived Neurotrophic Factor (*BDNF*), tyrosine receptor kinase B (*TRKB*) receptor promoter, (GABA)A receptor α1 subunit (*GABRA1*), Nuclear Receptor Subfamily 3 Group C Member 1 (*NR3C1*) gene, Corticotropin Releasing Hormone Binding Protein (*CRHBP*), FKBP Prolyl Isomerase 5 (*FKBP5*) and Neurotrophic Receptor Tyrosine Kinase 2 (*NTRK2*) were hypermethylated in the hippocampus of vulnerable suicide individuals, compared to controls, unlike the potential candidate gene psoriasis susceptibility 1 candidate 3 (*PSORS1C3*) which was hypomethylated across CpG sites of interest in the brain [47]. Serotonin (5-HT) expression is associated with the methylation of the 5-HT transporter gene (*SLC6A4*), and suicidality is associated with low *5-HT* expression and *SLC6A4* hypermethylation [51]. Additionally, rho guanine nucleotide exchange factor 38 (*ARHGEF38*) gene, linked to smoking cessation, showed hypomethylation across four CpG sites in BP patients who died by suicide [39]. Furthermore, regulation of GTPase and DNA methylation in *CACNA1C* (calcium channel subunit alpha 1 C) was linked to SAs. The disruption of the opioid signalling-pathway by hypermethylation of theoretically consequential DMRs may increase SB, emphasized after the discovery of the antidepressant-effect of Ketamine, requires the activation of the brain’s opioid-system, which if this latter was blocked or disrupted by increased methylation, would lead to an increase in SI precisely, especially in BP patients [39]. *SAT1* and spermine oxidase (*SMOX*) showed hypermethylation in suicidal samples [47]. Epigenetic mechanisms represent a promising avenue regarding SB therapeutics when it comes to suicide candidate genes. Modulating epigenetic modifications or the involved enzyme using properly designed specific molecules/inhibitors with high accuracy could help in many therapies as long as the side effects are within an acceptable range. 

#### 2.1.4. Other Genetic Endophenotypes Underlying SB

Other endophenotypes that can predict suicide in the long-run are represented by the impulsive/aggressive behaviours associated with the gene monoamine oxidase A (*MAOA*), which is possibly regulated by a long-non-coding RNA called MAALIN. *MAALIN* is found in a gene desert separating *MAOA* and *MAOB*, but specifically acts on *MAOA*. The homeostatic control of *MAALIN* expression is maintained by the distal promoter of the *MAALIN* gene that contains a repressive element which suppresses the transcriptional activity of the proximal promoter of the same gene. DNA methylation and histone modifications control *MAALIN* expression in a neuron-specific manner. Hypomethylation of *MAALIN* proximal promoter up-regulates *MAALIN* expression thus decreases MAOA expression as recorded in the dentate gyrus of individuals with obvious impulsive-aggressive traits who died by suicide. Long non-coding RNAs (lncRNAs) affect gene expression by several methods such as through direct DNA interactions, interaction with chromatin complexes, modulation of enzymatic activity, or interaction with DNA/RNA binding proteins. It is hypothesized that MAALIN regulates *MAOA* by serving as a scaffolding structure bringing chromatin remodelers into the *MAOA* promoter region, causing the switch from euchromatin to heterochromatin state repressing *MAOA* expression [52]. Moreover, the 5-methyl-cytosine (5mC), involved in neuropsychiatric disorders, is transformed into 5-hydroxymethyl cytosine (5hmc) (involved in brain plasticity, development, and disease). 550-CpGs were found to have suggestive differential hydroxy-methylation in depressed suicides [53]. Moreover, the gene body of Myosin XVI (*MYO16*) (involved in brain-development) revealed an increase in 5hmc and elevated expression in depressed-suicidal individuals, due to its location in an open-chromatin region accessible by chromatin marks. The gene body of insulin-degrading enzyme (IDE) that is expressed in insulin-responsive tissue showed increased 5hmc, and was downregulated (leading to an increase in insulin levels that in turn elevates the level of reactive oxygen species or ROS) in depressed suicidal individuals due to its presence in a region of weak transcription [53].

Eventually, the genetic-basis of suicidality varies with the variation of ethnicities, categories of traumas and mental diseases, highlighting the importance of the sample-size under study, SNP detection, methylation-patterns identification, and replication of studies to better understand the genetics of suicidality [28].

### 2.2. Neurobiological Underpinning of Suicidal Behaviour

#### 2.2.1. Structural Brain Changes in SB Patients

Neurobiology underlying SB requires further studies, although some limitations exist at this level due to hardships in assaying the brain. Suicidal completers show neurobiological abnormalities in the hypothalamic-pituitary-adrenal (HPA) axis and serotonergic-neurotransmission [54]. Alterations in brain structures, such as the frontal/prefrontal cortex, have also been found in people who died by suicide. These two regions are particularly involved in stress response, suppression of impulsiveness and cognition [55]. Suicide attempters’ diagnosis of the prefrontal areas showed alterations in the activation patterns, causing an impairment in social assessment, decision making and risk reward. The ventral prefrontal cortex (vPFC) showed lesser grey-matter volume and neural density in suicide attempters compared to non-suicidal patients with psychiatric disorders, when scanned by structural magnetic resonance imaging (MRI), emphasizing the role of the ventral and dorsal prefrontal cortex dysfunction in suicide [56]. While another study involving MRI scans of 11 MDD individuals with history of more lethal suicide attempts showed greater gray matter volume of the insula and prefrontal cortical regions, in comparison with lower lethality attempters and non-attempters MDD individuals, indicating a possible role of the PFC and insula in suicidal planning and intent; variations in results at this level may be due to heterogeneity of SB suggesting the existence of subtypes and confirmed by several assessment procedures, in addition to MRI, such as clinical assessment, positron emission tomography (PET) imaging, and ecological momentary assessments (EMA) [57,58]. Also, neuroimaging studies confirmed that high lethality attempters with MDD have relative hypofunction in the PFC compared to depressed low lethality attempters [59]. Furthermore, multimodal neuroimaging studies combining structural, functional MR imaging methods, PET imaging, and diffusion tensor imaging for adolescents and young adults with BP having or not having a history of SA, showed significant reductions in the gray matter volume in the bilateral cerebellum, right orbitofrontal cortex, and hippocampus of attempters compared to non-attempters; also in the uncinate and ventral and right cerebellum regions, a diminished integrity of the white matter was observed as well, in addition to a decreased functional connectivity between the amygdala, right rostral prefrontal and left ventral prefrontal regions (associated with severity of SI and attempt lethality, respectively, linking these regions to suicide-related symptoms and behaviours [60]). Significant brain structural changes were clearly seen in recent suicide attempters compared to other groups included in two separate studies then combined and compared in the third. Participants were patients of different suicidal backgrounds but all currently depressed. Study 1 (medication free) included remote suicide attempters, lifetime suicide ideators, and non-suicidal depressed patients. Study 2 (antidepressant treatment) included recent suicide attempters (within 3 days), lifetime suicide ideators, and remote suicide attempters. Study 3 represents pooled data from the two previous studies followed by comparison of the datasets and accompanied by correct imagining technologies to specific brain regions. Results showed that recent suicidal attempters compared to all other groups, have significant lower cortical thickness middle temporal gyri (associated with language processing, semantic memory, visual perception), smaller surface area of the insula (linked to impaired cognitive and decision-making processes associated with the transition from suicidal ideation to suicidal action.), smaller volume of the thalamus (critical role in relaying sensory input from subcortical regions and the centre of cognitive, emotional, and motivational processes that guide goal-directed behaviours), and larger volume of the nucleus accumbens (key region for action selection guided by frontal-cognitive and temporal-emotional inputs, modulate the decision and lethality for the suicide attempt). Furthermore, compared to remote suicide attempters, structural abnormalities in the limbic system in recent suicide attempters may point to the limbic system as a crucial element in the progression to SB by disrupting cognitive and behavioural processes (pain regulation, reward, and self-reference) driving susceptibilities for SB. Structural changes within two months of SB, and after a period of electroconvulsive therapy, and changes in cortical region thickness after a week of anti-depressant treatment proves the structural changes in a short period of time to be associated with SB [61]. Moreover, negative self-thinking and emotional stimuli processing are managed by the anterior cingulate cortex (ACC), which is highly activated in suicide attempters; also, the frontolimbic system in patients with high suicidal risk, has an abnormal resting-state functional connectivity [62]. Deep brain stimulation treatment for disorders displaying important extra-pyramidal symptoms (Parkinson disease), proved the correlation of SB and specific areas in the brain, through activating several neurobiological pathways due to off-target stimulation of some regions connected to ACC and amygdala (key neural substrates underlying emotional responses) or adjacent to substantia nigra (SNs) during the treatment. ACC activation results in disturbance of glutamatergic/GABAergic signalling, whereas SNs and dorsal striatum circuit activation causes dopaminergic dysfunction, increasing psychosocial distress and the incidence of SI [63]. Neurobiological research further indicates the involvement of the DRN in SB and other psychiatric conditions [17]. Recent neuro-imaging studies indicated that patients with SI and MDD have alterations in the connectivity of the characteristic brain-networks. The rostral middle frontal cortex, putamen, thalamus of the left hemisphere, caudate nucleus, frontal pole, pallidum, and superior parietal lobule showed reduced structural connectivity; whereas the frontal-thalamic circuit showed decreased functional connectivity [64].

Furthermore, the core-network in the brain connectome contains highly interconnected hub nodes called rich-club organization, and plays a vital role in the topology of brain networks and hierarchical sub-networks; it is found to be destroyed in individuals with depression [64]. In addition, the sub-frontal circuit involved in emotional processing, executive function and impulsivity, has a role in SI generation. The fusiform gyrus (FFG.L) that contains differential nodes might be a key component of network circuits reproducing SI, and seems to destroy the white-matter microstructure of the global network [64]. The disruption of neural circuits that control behaviour and influence emotions processing might be caused by alterations in the corticothalamic and thalamocortical pathways (frontothalamic loops), increasing the risk of suicidality in depressed patients [65]. Moreover, analysis of brain activity patterns revealed elevated amplitude of low-frequency fluctuations (ALFFs) resulting in an altered resting state brain activity in the hippocampus in samples with SI (related to how memories are processed [66]). Structural differences were also discovered during the examination of post-mortem ACC in suicidal individuals with depression at the level of gap junctions among glial cells (astrocytes, oligodendrocytes) that express and couple through different connexins. They showed a downregulation of the genes coding for connexins expressed by astrocytes. Other brain regions, for instance the DLPFC, showed downregulation of other markers alongside with connexins in suicidal depressed patients, accompanied by alterations in myelination as well (decrease in connexin-30 coupling to myelin-specific connexins) [67]. 

#### 2.2.2. Other Neurological Alterations

Samples from the post-mortem hippocampus of suicide completers showed an extreme downregulation of ErbB receptors (family of proteins containing four receptor tyrosine kinases), especially ErbB3, the receptor of neuregulin-1 (NRG1). ErbB3 is expressed in the neurogenic cells from division until functional maturity, and its decreased expression is accompanied with a decrease in the numbers of ErbB3-expressing granule cell neurons in the anterior dentate gyrus [68]. NRG1-ErbB3 signalling has an antidepressant-like effect and is involved in the increase in hippocampal neurogenesis. Chronic stress reduces the production of granule cell neurons in the anterior dentate gyrus resulting in the reduction of ErbB3, not related to differential methylation in the hippocampus, and this is considered as a novel suicidal endophenotype [68]. Genderwise, females are more susceptible to depression and SAs than males, indicating a possible role of female sex hormones (oestrogen and progesterone) in SB. Generally, oestrogen acts on neurons and glial cells by binding to the nuclear oestrogen receptors ERα and exhibit an antidepressant effect [69], proven by enhancing the antidepressant-effect of ketamine; In contrast, progesterone led to depression by binding to progesterone receptor membrane component 1 (PGRMC1) on microglia, lowering *BDNF* expression as reported [70]. Further assessment of female hormones’ depressive/anti-depressive effect is needed to clarify their exact mode of action in correlation with depression and SB.

### 2.3. Inflammation and SB

#### 2.3.1. Inflammatory and Immune Pathways

Inflammation’s role in suicide was proposed since 1993, when increased soluble interleukin-2 receptor concentrations were detected in suicide attempters, and it was later emphasized after extensive research and clinical evidence to uncover the relationship between the brain, imbalance of the immune system, and suicide pathophysiology [46,71]. Increased depressive symptoms with gender-variations were observed in the behaviour of samples who were injected with a low amount of the bacterial-endotoxin, lipopolysaccharide (LPS), due to alterations in cytokines concentrations measured in their plasma, and variations in inflammatory/neuroendocrine responses and negative emotional state [72]. Interleukin (IL)-6 cytokines are involved in SB. Besides macrophages and lymphocytes, IL-6 is also secreted by brain cells and binds to soluble IL-6R in the proinflammatory pathways used by brain cells impacting action potential, and thus influences emotions and behaviours affecting SB and other functions [46]. The presence of cytokines receptors on neurons verify their effect on neurons, thus influencing the aggressive/helplessness behaviours, by regulating the monoaminergic neurotransmitters and their metabolites in the CNS, in addition to synaptic transmission and plasticity regulation [73]. Furthermore, studies indicated that SB patients show increased levels of tumour necrosis factor-α (TNF-α) (remarkable increase in the DLPFC), transforming growth factor (TGF)-β1, vascular endothelial growth factor (VEGF), kynurenic acid (KYN), IL-1β, and IL-6 (high levels in the cerebrospinal fluid of suicide attempters), and lower levels of interferon (INF)-γ, IL-2, and IL-4 in specific brain regions [74]. Elevated levels of mRNA transcription of interleukins IL-4 and IL-13 were observed in the orbitofrontal-cortical area in suicidal deaths also [75]. Peripheral inflammatory markers can distinguish between depressed suicidal and depressed non-suicidal patients, or even between suicide attempters and non-suicidal depressed according to the levels interleukins, mainly IL-2 and IL-6, TNF-α S100 and C-reactive protein [76,77]. Classical monocytes were also activated in the blood samples of patients with SB, suggesting the possible role of adaptive/innate immune system in affecting mood and SB [78]. Likewise, vitamin-D shifts T helper balance towards Th2 phenotype leading to a less pro-inflammatory state, indicating that vitamin-D deficiency is linked to higher inflammation possibility and higher risk of suicide [79]. Patients that suffer from autoimmune disorders such as multiple sclerosis (MS), celiac disease and lupus erythematous have higher rates of suicidality compared to other diseases. Autoimmunity, stress, traumatic brain injury (TBI) and neurotrophic pathogens are mechanisms that trigger inflammation and involve changes in emotions and behaviour in suicidal subjects. Notably, immunotherapy with IFN-α or β in patients with cancer or MS, may lead to the development of depression, SI and SAs later [80]. In addition, asthma and allergy cause increased upper airway inflammation and obstruction which may lead to depression and stress. Epidemiological studies showed a two-fold to three-fold increase in SB in patients with asthma and allergy compared to the control group [81]. 

#### 2.3.2. COVID-19 Pandemic and SB

Recently, studies showed the impact of SARS-CoV-2 on certain biological pathways such as the renin-angiotensin systems, nicotine receptors and central/systemic inflammation, that overlap with SB triggering [82]. The spike protein of SARS-CoV-2 interacts with the angiotensin-converting enzyme 2 receptors of host cells that are highly expressed in human cells including CNS cells. The infection-related brain damage involves multiple mechanisms: peripheral inflammation that modulates brain function, retrograde axonal transport of the virus from the respiratory system and migration of mononuclear cells transporting the virus across the blood-brain barrier. The recent pandemic has a social/emotional influence and respiratory complications, making the fear of contagion, stress, guilt, loss of routine, pain, insomnia, anxiety, loneliness and obligation of social-distancing/quarantine—social connectedness is critical to social and emotional stability and disengagement increases suicidal risk rate—more threatening, thus leading to depression and the development of other mental disorders elevating the risk of SB in a vulnerable population [82]. In comparison to pre-pandemic studies, a meta-analysis across 54 studies (308,596 participants) detected an increase in the rate of SI (10.81%), self-harm (9.63%) and SA (4.68%) during COVID-19 pandemic, but larger samples are required to have a stable estimate regarding the COVID-19/suicidality link in the general population. The negative impact of COVID-19 varies considerably among people and places. This meta-analysis revealed that health measures, media and political systems determine how SB manifests during the pandemic indicating. For instance, the most vulnerable to SI are young people, women and people from democratic countries [83]. COVID-19-related suicide cases from the general population or among healthcare workers were reported in many countries (Italy, Germany, UK, US, India and others) of varying age groups and vulnerability depending on many factors (area of prevalence of COVID-19, loss of a close relative due to infection, etc.,) [84]. Moreover, mental health problems were reported among social and healthcare workers during the pandemic indicating an elevated risk of suicides particularly among frontline workers, with variations among countries. Data revealed that in China, for instance, no STBs prevalence between front-line and non-frontline neither between healthcare workers nor the general population. Similarly, in the USA, there was no significant SB when comparing between healthcare workers and other groups. In Mexico there were higher levels of STBs in frontline healthcare workers compared to non-frontline. On the other hand, in Belgium, hospitalization of healthcare workers for being infected with COVID-19 showed a significant association with STBs, which was absent in Spain in a larger cohort study [85]. A primary analysis of 21 countries (6 high-income and five upper-middle-income countries) to study suicide trends during the COVID-19 pandemic has showed either a decrease or no significant change susceptible to variations between areas in the same country. For example, in Thames valley, England, UK there was a significant decrease in suicide was reported; similar results were found in other countries such as Alberta and British Columbia in Canada, California, Illinois and Texas in USA, and New South Wales in Australia. Conversely, Puerto Rico and New Jersey (USA) showed an increase in suicides during the pandemic’s early months [86]. Moreover, approximately a 16% relative increase in suicidal rates and an escalation of SI in young adults were reported during the COVID-19 pandemic in Japanese and Australian populations, respectively [87,88]. Robust studies are required in this field to emphasize the clinical evidence and confirm observations among comparisons and to take suitable approaches or measures to improve mental health and decrease suicidal risk during pandemics.

#### 2.3.3. Neuroinflammation and SB

The hypothalamic-pituitary-adrenal (HPA) axis has an endocrine-function and is activated by proinflammatory cytokines (Figure 2) released after environmental, emotional/physical stress [89,90]. In addition, childhood adversities and traumatic events contribute to HPA-axis hyperactivity/dysregulation in adulthood, increasing the risk of developing a psychiatric disorder and/or attempting suicide [28]. Activated HPA releases the corticotropin-releasing hormone (CRH) and arginine-vasopressin from the hypothalamic-para-ventricular nucleus, that interact with their receptors on the pituitary gland leading to the release of adrenocorticotrophic hormone (ACTH), stimulating the secretion of glucocorticoids (GC or cortisol in humans) from the adrenal cortex [91]. Cortisol has a regulatory function (regulates immunity, neuronal survival and genesis, formation of new memories; also causes atrophy in hippocampus and amygdala that are linked to psychopathological disorders) on all body areas including CNS, and, in normal conditions, it induces negative-feedback on the HPA [92]. The consecutive release of proinflammatory cytokines causes cortisol resistance and impairs the negative feedback resulting in HPA-hyperactivity. Increased cortisol levels activate microglia and induce neuroinflammation, impairing BDNF function then causing neurotoxicity resulting in neuronal cell death [93]. Dysregulation of the HPA axis occurs due to a decrease in the GC receptors’ sensitivity in the hippocampus and anterior pituitary caused by polymorphisms in the FKBP Prolyl Isomerase 5 (*FKBP5*) gene that regulates and influences the GC receptors’ sensitivity [94]. SB could be related to direct or down-stream CRHR1 actions. CRHR1 polymorphisms and childhood traumas have an impact on decision-making in suicide attempters [28]. HPA dysregulation is also associated with other pathways involving the opioids -a neuropeptide involved in regulation of emotions and other functions [95], 5-HT, lipid status, neurogenesis/neuroplasticity, neuroinflammatory pathways and glutamate systems that also play a role in suicidality [23]. Among these divergent systems impaired by neuroinflammation, the serotonin system historically plays an important role confirmed by PET imaging and EMA [96]. Serotonergic abnormalities including changes in the 5-HT_2A_ receptor binding potential are related to SB and aggression [59]. The DRN from which originates 5-HT innervation in the brain, modulates the activity of several brain regions including the PFC [97]. Suicidal individuals’ samples revealed changes at the level of rDNA transcriptional activity of the DRN neurons. This might be due to immune activation/neuroinflammation whose impact is deteriorating on these neurons [97]. Microglia (5–10% of brain cells) are resident macrophages in the CNS. They represent the brain’s first line of defence and produce nitric oxide and cytokines [98]. Depressed suicidal individuals showed an elevation in the level of activated microglia in post-mortem brain white-matter [99]. Once activated, microglial cells are polarized towards one of two phenotypes (Figure 2), M1 which is proinflammatory and M2 which is anti-inflammatory [100]. NADPH oxidase 2 (NOX2) elevation was associated with the increase in IL-6 levels [101]. Abnormal microglial activation plays a key role in SB; Compared to non-suicidal depressed patients, suicidal patients revealed an elevated microglial activity inducing oxidative stress, that in turn causes a decrease in the rDNA transcription activity in DRN neurons leading to neuroplasticity deterioration [97]. The increase in the Human Leukocyte Antigen –DR isotype- presentation in microglial cells, confirms the cross-talk between microglia and neurons. The decreased microglial activity has a restorative function in non-suicidal samples indicating how devastating the neurodegenerative role of microglia in suicide can be [97]. Another indication of neuroinflammation is the significant increase in the level of translocator protein (TSPO) in activated microglial cells in the ACC and the insula in patients who experienced SI compared to those who did not [94].

Neuroinflammation stimulates the kynurenine pathway (KP) leading to melatonin and serotonin depletion. Aggression and impulsivity are greatly associated with the decrease in 5-HT levels [102,103]. Tryptophan is the precursor of serotonin, and 90% of this amino acid is broken down through KP due to neuroinflammation and excitatory amino-acids, dropping the levels of 5-HT in the brain [104,105]. IFN-γ, IL-1β, and IL-6 stimulate indoleamine-2,3-dioxygenase (IDO), the enzyme responsible for degradation of tryptophan into N-formyl kynurenine, which in turn is degraded into several metabolites such as kynurenine, quinolinic acid (QUIN) -an N-methyl-D-aspartate (NMDA) receptor agonist- kynurenic acid (KYNA) and picolinic acid (PIC) [104]. These KP metabolites strongly affect mood and behaviour throughout several mechanisms including the modulation of the glutamate neurotransmission and neuroinflammation [104,106,107]. The elevation of QUIN level has a neurotoxic-effect and causes an overreaction in the glutamatergic system and reduction in BDNF production, resulting in cognitive impairment and worsening of neuroplasticity [108]. The elevation in QUIN and KYNA levels have a consequence on the depletion of 5-HT and melatonin. An increased level of kynurenine metabolites and cytokines (specifically IL-6) in CNS and blood are detected in suicidal patients [14,109,110,111]. A 300% increase in QUIN levels were observed in the CSF of suicide attempters regardless if they had a mental disorder or not [112]. Moreover, the kynurenine level was higher in suicide attempters compared to depressed patients. Conversely, PIC is neuroprotective and its level was reduced in the CSF and blood, and the PIC/QUIN ratio was decreased in the CSF and blood of patients with SB [113]. A detailed description of the involvement of KP in suicide is out of the scope of this review and can be found elsewhere [114,115,116]. BDNF, a neurotrophin family member, is a GF implicated in the proliferation, differentiation, migration, survival of CNS neurons, regulation of synaptic activity and maintenance of neural plasticity, and it is regulated by stress and 5-HT mechanisms [117]. The signalling cascade is initiated when this GF binds to tyrosine kinase B (TrkB). Studies showed a decrease in the protein and mRNA levels of BDNF in the PFC of young suicidal individuals. Whereas in the hippocampus, the decrease was observed at the level of the BDNF mRNA level only; but, the BDNF receptor (TrkB) protein and mRNA levels were reduced in both the PFC and the hippocampus [117]. The association between mood disorders and abnormalities in both neuroplasticity and neural atrophy emphasizes the role of the BDNF in suicide pathophysiology. Additionally, BDNF/TrkB signalling abnormalities can result in brain structure deformity in suicidal patients due to its effect on apoptotic pathways [117].

Inflammatory changes in brain parenchyma and increased inflammatory metabolites and proinflammatory cytokines levels, caused by TBI, is associated with a higher risk of SB. TBI patients display higher levels of TNF-α than controls, related to SI progression [118]. Additionally, brain infections by *Toxoplasma gondii* cause GABA pathway/dopamine dysregulation in infected patients, rendering them at higher SB risk [119]. *T. gondii* infecting patients with behavioural disorders could possibly induce a “behavioural manipulation” in the host (e.g., reduction of fear, as revealed in animals with induced infection with *T. gondii*). This affects the host transcription elements through epigenetic modifications and induction of special effectors (such as the *Toxoplasma* E2F4-associated EZH2-inducing Gene Regulator), resulting in a change in the host gene expression at both transcriptional and post-translational levels [120].

### 2.4. Metabolism 

The post-mortem brains of suicidal individuals showed noticeable biochemical changes in NADPH oxidase NOX2 enzyme, expressed in the CNS, a protein that transfers electrons across biological membranes producing superoxide [16]. NOX2 is a major source of ROS (Figure 3) in pathological conditions, such as psychiatric disorders, neurodegenerative diseases and SB [16,101]. 

Thus, oxidative-stress in the brain plays a role in suicidality, and to further understand NOX2’s role in suicidal deaths, NOX2 expression in cellular brain subpopulations (neurons, microglia and astrocytes) was investigated [101,121] by immunohistochemical analysis of NOX2 focusing on the cortex of suicidal samples. Double immunohistochemistry precisely evaluated that NOX2 expression was mainly increased in cortical neurons of suicidal samples, more specifically in cortical GABAergic neurons. In brief, NOX2 embodies a potential biomarker for suicide prediction. In addition, the mitochondria-derived ROS is another source of oxidative-stress. The consumption of special nutrients such as vitamin C, zinc, vitamin B12, folic acid, w-3 fatty acids [101] and magnesium, could be of considerable utility in preventing SBs and mood disorders in vulnerable people and specific age groups due to their role in preventing mitochondria-derived-ROS production. Notably, brain oxidative damage caused by sleep deprivation may be related to suicidality [16,101].

Mitochondrial and membrane lipids functions can be damaged by severe stress affecting the downstream processes, neurotransmitter signalling and information processing in synapses and circuits (motoric, neurovegetative, cognitive, affective behaviours) [16]. Higher risk of SAs and suicide were also related to two dietary lipid classes: cholesterol and polyunsaturated fatty acids (PUFAs) [8]. Together with sphingolipids form the so-called lipid-raft-region, a critical region of membrane lipids in the CNS mediating the neurotransmitter signalling through G-protein coupled receptors and ion channels [16]. Low cholesterol and elevated n-6 to n-3 PUFAs have been associated with an increase in suicidal risk [8].

## 3. Pharmacotherapy

Treatment of SB requires joined efforts and a wide range of research and possibilities, and in spite of the excessive data available, pharmacotherapy of SB is not the main focus of this review and will be discussed in brief. Treatment options, in turn, are dependent on regulations and allowances of each country [122]. Among pharmacological treatment options, ketamine demonstrated high efficacy in reducing suicidality short-term after intravenous infusion, although long-term effects still need to be clarified [123]. Approved therapeutics include several drugs administered to depressed patients to decrease the possibility of SI and SA, such as Lithium (mood-stabilizing drug), Clozapine (reduces suicidality, aggressive/impulsive behaviours), and Ketamine/Esketamine (antidepressant) [124]. An Australian study also showed the critical role of general practitioners in preventing suicide [125]. Drugs, social efforts, psychological awareness and guidance should be accompanied with a healthy lifestyle and integrated nutrition to resist SB (Figure 4).

## 4. Conclusions and Future Perspectives

In summary, there is no single aspect behind suicidal behaviour and triggering factors vary widely among age groups, countries, areas within the same country, regions with cultural/social and economic differences, and even between individuals according to the available combination of stressors, vulnerability and other risk factors. Based on data collected from recent studies examining the diverse factors, triggers and pathways implicated in SB, this review confirms that SB is a complicated, multifactorial, polygenic and independent mental disorder, yet an interesting field of research. Suicide, a major global health concern and a leading cause of death among young people particularly, is caused by a cascade of interactions, reactions and alterations that when combined may pave the way towards SB. The complexity in decoding SB is due to the interconnected basis, the branched probabilities and the major limitations against it, rendering research through this phenomenon a very convoluted journey to disentangle the real combination of causes leading to suicidality.

Future studies examining novel data and replication studies for existing findings of populations with diverse genetic makeup, variable environments/social experiences, association with mental disorders, and altered neuroinflammatory pathways would navigate the possibility of clarifying the still many unanswered questions, opening the gate towards effective therapeutic procedures through the discovery of novel biomarkers that can be of great importance regarding the upstream triggers of SB, and the downstream effectors.

The understanding of SB represents the foundation to cure/prevent it. Given that the different underpinnings of suicide described here including neuroinflammation, engaged inflammatory pathways, alterations in specific brain regions, altered regulation of epigenetic mechanisms, malnutrition, social experiences, childhood traumas, social awareness, psychological factors, and relative mental vulnerability, it is very likely that the field will have to develop toward precision medicine and targeted therapy rather than a one-drug-fits-all model. Remarkably, the improvement of molecular techniques/technologies in neuroimaging and scanning is indispensable to uncover the neural networks and their molecular/biochemical communication, as well as their protective strategies against harmful triggers. Moreover, deciphering the mystery beyond cognition, emotions, behaviour, mood, and memory can by itself be an extraordinary revolution in neuroscience, towards gaining novel knowledge about brain functioning. Future research concerned with thoughts dissection, collection, storage and memory building will help researchers navigate predictable suicides. Thus, suicide requires multidimensional strategies, databases and efforts to be understood, attenuated and most importantly, prevented.

## Figures and Tables

**Figure 1 brainsci-13-00505-f001:**
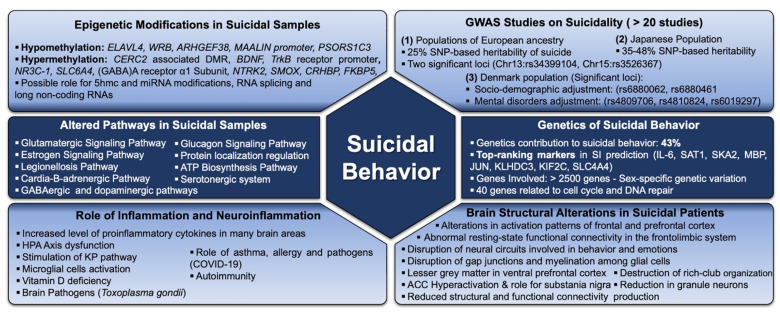
General summary of the genetic and neurobiological factors underlying suicidal behaviour. (Created in BioRender.com (accessed on 2 April 2022)).

**Figure 2 brainsci-13-00505-f002:**
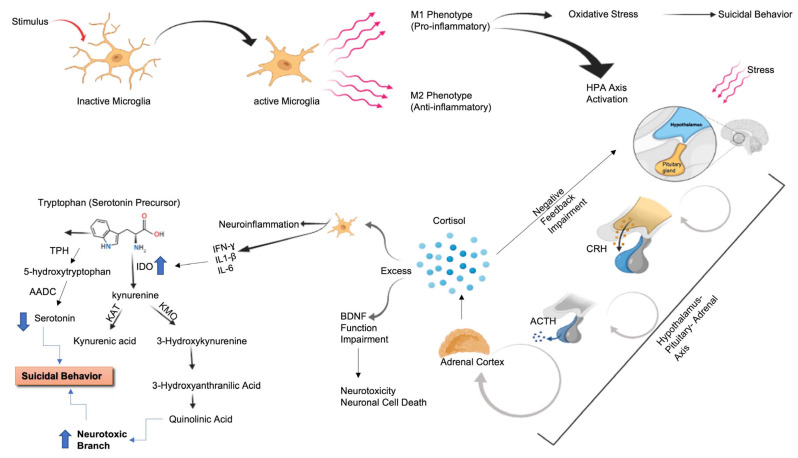
Role of microglia and hypothalamic-pituitary-adrenal-axis in neuroinflammation and suicidality. Emphasis on the kynurenine pathway. (Created in BioRender.com (accessed on 2 April 2022)).

**Figure 3 brainsci-13-00505-f003:**
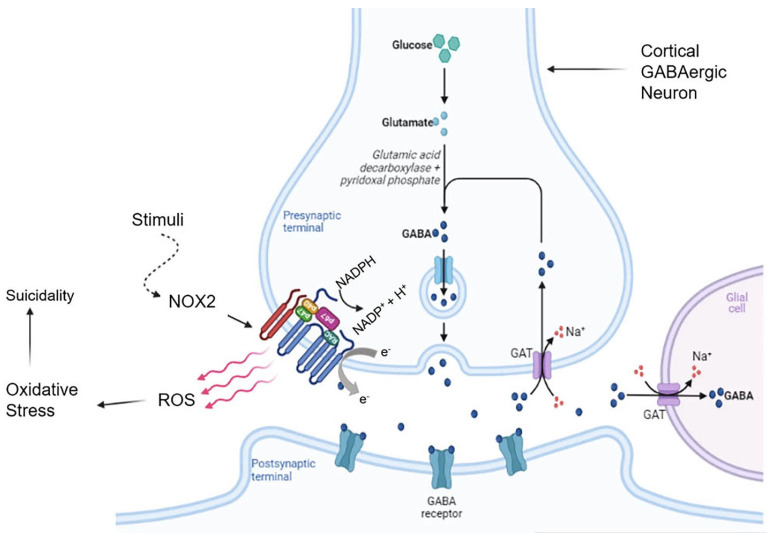
Oxidative Stress and Suicidality: Role of NADPH oxidase 2 in Reactive Oxygen Species production. (Created in BioRender.com (accessed on 2 April 2022)).

**Figure 4 brainsci-13-00505-f004:**
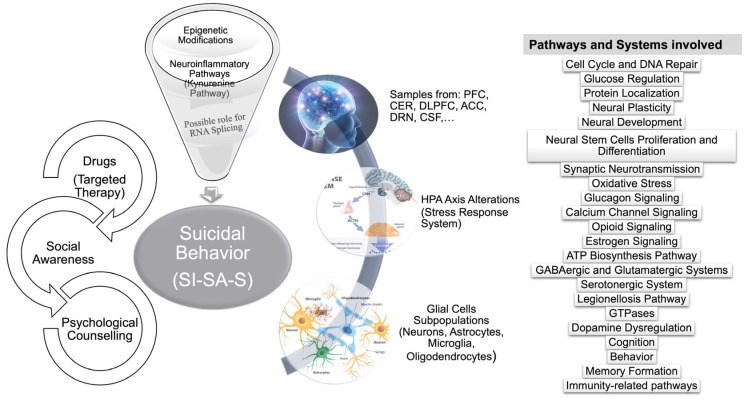
Aspects of suicidal behaviour: phenotypes, altered brain regions, pathways/systems involved, and therapeutic procedures.

## Data Availability

The data presented in this review are available within the article text and figures.

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
