# Peer review of "Biological Factors Underpinning Suicidal Behaviour: An Update"

_brainsci, 2023, doi:10.3390/brainsci13030505_

Round 1

Reviewer 1 Report (Previous Reviewer 2)

I mentioned some fundamental methodological issues that can not be addressed during revisions. Adding databases is a real worrisome during a revision. Still, the paper highlights biological aspects, already established facts, narrative review, and questionable search data bases. 

Author Response

Reviewer 2 Report (Previous Reviewer 1)

The authors have adequately addressed my concerns.  

Author Response

Reviewer 3 Report (New Reviewer)

This study reviewed the evidence about various biological factors involved directly or indirectly in the pathophysiology of suicidal behaviors. Most of the studies reviewed in this article are published in recent 3 years. I believe that this article can help readers increase knowledge of suicidal behaviors.

The authors may consider make some minor revisions to improve their manuscript.

1.      The title can be revised to emphasize the “Biological Factors Underpinning Suicidal Behaviour.”

2.      As the authors stated in Introduction, the etiologies of suicidal behaviors are multi-factorial. The authors may consider add a short paragraph to introduce the importance of understanding biological factors of suicidal behaviors but not only focus on psychosocial factors.

3.      Line 31 “mental disorder” can be changed into “mental health problem.”

Round 2

Reviewer 1 Report (Previous Reviewer 2)

I mentioned the flaws which would not be addressed during the revisions, I assume. 

This manuscript is a resubmission of an earlier submission. The following is a list of the peer review reports and author responses from that submission.

Round 1

Reviewer 1 Report

The authors summarise the existing literature surrounding suicidal behaviour. Overall a reasonable piece of work, however the manuscript could be improved by restructuring and by attempting to tie the different strands of evidence together, for example, do the different lines of evidence demonstrate any convergence?

Major comments

·         Page 2, line 48: “the intentional injury arising from purposeful actions directed towards oneself…” Many previous studies include deliberate self-harm on the spectrum of suicidal behaviour, as it is a significant risk factor for later suicide attempt and/or can escalate to include accidental self-inflicted death. My understanding is that it is related but incompletely overlapping with suicidal thoughts and behaviours. Please be more clear on how/where self-harm is considered in this article.

·         In the introduction the authors list a number of risk factors for suicidal behaviour, but do not mention access to lethal methods. This is an important factor and should be included.

·         The section on genetic/molecular factors needs revising for clarity, specifically the terminology needs to be corrected here. locus/loci are regions of DNA. A SNP is a single base pair change. A gene is a region of DNA that encodes a protein/mRNA entity. These terms cannot be used interchangeably. A GWAS is a genome-wide association study, not to be confused/used interchangeably with an epigenome-wide association study (page 3 line 125).

·         Please also be specific about the population (clinical or general population?), phenotype assessed in each GWAS, as well as the covariates. What is meant by “socio-demographics” and how does this adjustment different from the standard age, sex and population structure covariates used for standard genetics studies? Similarly, the sample size of the study is important: did the Japanese study find no significant variants because it was much smaller that the European ancestry studies? which of these findings can be considered robust (big sample sizes and/or replication/validation)?

·         Restructuring/more subheadings could be used to increase clarity/logic flow of the manuscript.

·         Restructuring the section on genetic/molecular factors could be clearer. Currently genes/genetic variants/methylation studies/findings are all muddled up together.

·         The review of genetic studies is incomplete, with a number of studies being omitted: PMID: 36515925, 35347246, 34861974 

·         More specificity is required, for example page 4, line 153: “alterations in the cannabinoid receptors…” alterations in expression level? Or the sequence? Or binding? Similarly page 5 line 171 “changes have been recognised…”

·         Regarding evidence of brain structure and function: given that suicidal symptoms fluctuate over time (from minutes/hours to months/years), it is possible to see how changes in connectivity might be associated with suicidal behaviour (as connections fluctuate, but it is less clear how the structural changes (which are less variable over short time periods) might be associated. Please clarify this point.

·         The authors highlight an increase in SB in 2 populations during the pandemic (page 8, line349), but other countries demonstrated no increase (for example the UK). A more balanced view is required (rather than picking only the reports that fit the hypothesis).

Minor comments

·         Is suicide victim the best phrasing? Suicidal individual would be my preference (however I don’t know if there is a consensus on this terminology).

·         Adding gene symbols in the text, in addition to the long names, would make reading the gene-based summary easier (such as page 3, line 126). Consistency in used of names vs gene symbols would be good (for examepl page 5 line166)

·         Gene names should be presented in italics consistently. This is particularly important for differentiating protein and gene expression levels (page 3, line 145 onwards).

·         Page 3, line102. “genetic/epigenetic mechanisms represent promising avenues regarding SB therapeutics”. How would genetic/epigenetics be modulated in a therapeutic setting? This seems to be rather a stretch.

·         Page 12, line460: if NOX2 gene and protein expression changes are observed in neurons, how does this translate to utility as a biomarker? Brain tissue is not easy to assay.

·         There are some somewhat unexpected jumps in logic that should be avoided/better explained. For example, page 8 line 339: “Suicide is related to asthma and allergy due to increased upper airway inflammation and thus depression”. If upper airway inflammation were that obviously linked to depression more cases of depression could be treated/prevented. Possibly this is a clumsy interpretation of the paper, however accuracy is needed.

·         Consider splitting some sentences for clarity, as some are very long. For example page 13, line 504.

Reviewer 2 Report

Thanks for sending the paper. I found that the author reviewed the already available evidence from database PubMed which raises significant concerns about novelty. The authors mentioned that the SB is multifactorial, however, they discussed only biological aspects, completely ignoring the psychosocial aspects. It is important to remember that no single aspect can explain suicide to date. 

Weakness- narrative review, single database searched, only biological aspects highlights  
